**Data Availability Statement:** All relevant data are within the paper and its Supporting Information files.

# Willingness to receive COVID-19 vaccine and associated factors among adult chronic patients. A cross-sectional study in Northwest Ethiopia

**Masresha Derese Tegegne**[1☯]*, **Surafel Girma**[2☯], **Surafel Mengistu**[2☯], **Tadele Mesfin**[2☯], **Tenanew Adugna**[2☯], **Mehretie Kokeb**[3☯], **Endalkachew Belayneh Melese**[4], **Yilkal Belete Worku**[4], **Sisay Maru Wubante**[1]

1 Department of Health Informatics, Institute of Public Health, College of Medicine and Health Sciences, University of Gondar, Gondar, Ethiopia, 2 Department of Medicine, College of Medicine and Health Sciences, University of Gondar, Gondar, Ethiopia, 3 Department of Pediatrics and Child Health, College of Medicine and Health Sciences, University of Gondar, Gondar, Ethiopia, 4 Department of Internal Medicine, School of Medicine, College of Medicine and Health Sciences, University of Gondar, Gondar, Ethiopia

☯ These authors contributed equally to this work.

\* masresha1derese@gmail.com

## Abstract

### Background

People with pre-existing chronic diseases are more likely to acquire COVID-19 infections, which can be fatal, and die from COVID-19 illness. COVID-19 vaccination will benefit those at a higher risk of developing complications and dying from the disease. This study aimed to determine chronic patients' willingness to receive a COVID-19 vaccine and the factors that influence their willingness.

### Method

An institutional-based cross-sectional study was conducted among 423 adult chronic patients in the University of Gondar specialized hospital outpatient departments. The participants were chosen using systematic random sampling methods with an interval of 5. Face-to-face interviews were used to collect data from eligible respondents. Epi-data version 4.6 and SPSS version 25 were used for the data entry data analysis. Bivariable and multivariable binary logistic regression analyses were used to evaluate the relationship between the dependent and independent factors. An odds ratio with 95 percent confidence intervals and a P-value was used to determine the association's strength and statistical significance.

### Result

Out of 401 respondents, 219 (54.6%) with [95% CI (49.7–59.5%)] of study participants were willing to receive the COVID-19 vaccination. Being a healthcare worker (AOR = 2.94, 95% CI: 1.24–6.96), Lost family members or friends due to COVID-19 (AOR = 2.47, 95% CI: 1.21–5.00), good knowledge about COVID-19 vaccine (AOR = 2.44, 95% CI: 1.37–4.33),

**Funding:** The author(s) received no specific funding for this work.

**Competing interests:** The authors have declared that no competing interests exist.

**Abbreviations:** AOR, Adjusted odds ratio; CI, Confidence interval; Epi-info, Epidemiological Information; -19, Coronavirus diseases 2019; COVAX, COVID-19 Vaccines Global Access Facility, SPSS: Statistical Package for Social Science.

favorable attitude towards COVID-19 vaccine (AOR = .8.56 95% CI: 4.76–15.38), perceived suitability of the COVID-19 infection (AOR = 2.94, 95% CI: 1.62–5.33) and perceived benefit of the COVID-19 vaccine (AOR = 1.89, 95% CI: 1.08–3.31), were found to be a significant association with the willingness to receive the COVID-19 vaccine among chronic patients.

## Conclusion

This study confirms that around 55% of adult chronic patients were willing to receive the COVID-19 vaccine. Providing health education for chronic patients to emphasize the knowledge and attitude of the COVID-19 vaccine and raise patients' perceived risk of COVID-19 and the benefit of the COVID-19 vaccine could be recommended to improve their willingness to COVID-19 vaccination.

## Background

The COVID-19 pandemic, also known as the coronavirus pandemic, is a global health threat caused by SARS-CoV-2 virus [1]. Since its outbreak in December 2019, it has triggered a global catastrophe that has had a devastating influence on the worldwide community and people's lives, health, livelihoods, economies, and behaviors [2–6]. There were more than 520 million cases of COVID-19 worldwide as of May 14, 2022, resulting in 6.2 million deaths [7, 8].

COVID-19, on the other hand, does not affect everyone in the same way. People who have pre-existing chronic conditions, such as hypertension, diabetes, HIV/AIDS, cardiovascular disease, cerebrovascular disease, or pulmonary disease, are more likely to develop severe infections and die from COVID-19 illness [9]. This is because an underlying sickness inhibits the immune system, protecting our body against diseases [9].

The COVID-19 Vaccines Global Access (COVAX) facility aims to provide at least 2 billion vaccine doses to concerned countries by 2021, with donors contributing at least 1.3 billion doses for the 92 low-income countries [10]. COVID-19 vaccines were manufactured within one year after the WHO declared COVID-19 an international public health emergency. Due to remarkable vaccine research, development, and production determination, COVID-19 vaccines were developed within the shortest period of vaccine production history [11, 12]. Astra-Zeneca vaccinations manufactured by the Serum Institute of India (SII) were supplied to Ethiopia on March 7, 2021, as part of the steps taken to fight a recent increase in COVID-19 infections [13]. Ethiopia got approximately 2,184,000 doses of COVID-19 vaccine from the COVAX facility out of 7,620,000 doses. [14, 15].

COVID-19 vaccinations, like all vaccines, are chemicals that stimulate antibody production and provide protection against COVID-19 diseases. As a result, vaccinated people have a lower risk of being sick, spreading the virus to others, becoming extremely ill, and requiring hospitalization. If a person gets immunized, the chances of suffering short- and long-term problems are reduced [16, 17]. Furthermore, the disease's socioeconomic burden is reduced [17].

Even while everyone benefits from the vaccination, not everyone benefits equally. Those at a higher risk of contracting COVID-19 infection, more likely to get very unwell once infected, or at a higher risk of developing complications and dying from the disease will benefit the most from vaccination [18, 19]. Patients with chronic conditions such as diabetes, hypertension, cardiovascular, renal, and cerebrovascular diseases were given priority for vaccination

since they were more likely to be affected by this virus and have a significant clinical burden from COVID-19 [18, 20].

Vaccination campaigns have begun in various countries using different implementation strategies depending on availability, roll-out speed, and willingness rates. The global population's willingness to get the COVID-19 vaccine is relatively high [21]. On the other hand, refusing to receive the vaccine might hinder international efforts to control the present outbreak [22]. According to the systematic study conducted in each region, the main reasons for vaccine hesitancy were fear of side effects and unpleasant reactions [23, 24]. Furthermore, this is facing a significant obstacle as more and more people are becoming unwilling to be vaccinated, hindering the efforts to control the spread of the disease [15]. Because of the severity of the problem, the WHO now considers vaccination hesitancy to be a serious threat to global health [25, 26].

There is no data relevant to people with chronic diseases in our context, Ethiopia; the Ministry of Health's goal is to vaccinate 20% of the population by the end of 2021 [27], but as of October 14, 2021, only 2.6 percent of the overall population has gotten at least one dose of the vaccine. Therefore, research into chronic patients' willingness to receive the COVID-19 vaccine is essential for creating awareness. Any country with a vaccination gap is likely to see the emergence and spread of new, difficult-to-treat strains [28]. As a result, research is needed to understand the chronic patients' willingness to receive a COVID-19 vaccine and the factors that influence their willingness.

## Methods

### Study design and setting

An Institutional based cross-sectional study was conducted among chronic patients from November to December 2021. The University of Gondar comprehensive specialized hospital is located in Gondar City in Amhara regional state. Gondar city is situated 748 km northwest of Addis Ababa, the capital city of Ethiopia. Currently, the city has one Referral Hospital and eight government Health Centers. University of Gondar Comprehensive specialized hospital provides a full range of health care services, including outpatient, inpatient, and surgical services. The hospital has 1040 health care professionals, 580 beds in five different inpatient departments and 14 wards, 14 different units giving outpatient services to customers, and around 800 patients are seen in the hospital every day. Furthermore, the outpatient department serves approximately 6000 chronically ill patients who are followed up on a regular basis based on their unique disease and severity of illness. In addition to the health care services, the hospital also serves as a site of knowledge transfer and development by providing teaching and research activities serving more than seven million people of northwest Ethiopia and peoples of the neighboring zones and regions [29].

### Study populations and eligibility criteria

Patients with chronic diseases such as chronic liver disease, chronic kidney disease, hypertension, diabetes mellitus, and heart failure were included in this study. Chronic disease was diagnosed using clinical features, laboratory findings, and imaging results. Furthermore, when serum IgG levels are relatively high, it is referred to as chronic illness [30]. The treating physicians determined the diagnosis based on the criteria given above. This study comprised patients who had their follow-up at the University of Gondar specialized hospital outpatient department. However, patients who were mentally ill, unable to speak, and had a major hearing problem were excluded from the study. This is because patients who are mentally ill,

disabled, or unable to hear or speak were unable to respond to inquiries concerning their willingness and permission.

## Sample size

We used the single population proportion formula to determine the sample size, $n = Z (\alpha/2)^2 pq/d^2$ [31]. We assumed n = the required sample size, Z = the value of standard normal distribution corresponding to $\alpha/2$ = 1.96 (with 95% confidence level), p = prevalence of patients who were willing to take the COVID-19 vaccine, q = prevalence of patients who were not willing to take the COVID-19 vaccine (1-p), d = precision assumed as (0.05). We used a p-value of 0.5 because there had been no previous research on chronic patients' willingness to take the COVID-19 vaccine in Ethiopia. Hence, the calculated sample size was to be 384. After considering a 10% non-response rate, our study subjects will be 423 people with chronic diseases.

## Sampling procedure

A systematic random sampling technique was used to choose study participants. Every day, an average of 150 chronic patients visit the outpatient departments of the University of Gondar specialized Hospital. A total of 2,250 chronic patients were expected during the data collection period (15 working days). By dividing the total population by the sample size (423), the sampling interval was found to be 5. The lottery method was utilized in the first interval, and every fifth chronic patient was picked in the following intervals until we obtained a total of 423 samples.

## Study variables

The primary outcome variable of this study was the willingness to take the COVID-19 vaccine among chronic patients. Participants' demographic characteristics, current knowledge about the COVID-19 vaccine, attitude toward the vaccine, and perceptions of their susceptibility to COVID-19, the COVID-19 vaccine's benefit, and potential vaccine harms were the explanatory variables included in this study.

## Operational definitions

*Willingness to take the COVID-19 vaccine* was measured using the question "I am willing to be vaccinated against COVID-19" by using a 5-point Likert scale from 'Strongly agree' (score 5) to 'Strongly disagree' (score 1) (Strongly Agree, Agree, Neutral, Disagree, Strongly disagree) [32]. The willingness variable was further dichotomized to "willing to take COVID-19 vaccine" (1 = agree/strongly agree) and "not willing to take COVID-19 vaccine" (0 = strongly disagree/disagree/neutral) [33].

*Knowledge about the COVID-19 vaccine* was measured using six items of questions [15]. Those who answered "Yes" to the knowledge questions received 1 point, while a "No" answer was given 0 points. Respondents who scored 70%(4.2–6) or higher on the knowledge question were judged to have good knowledge of the COVID-19 vaccination, while those who scored less than 70%(4.2) were regarded to have poor knowledge of the vaccine [34].

*Attitudes towards the COVID-19 vaccine* were assessed by 9 item questions on a 5-point Likert scale from 'Strongly agree' (score 5) to 'Strongly disagree' (score 1)(Strongly Agree, Agree, Neutral, Disagree, Strongly disagree) [15]. The respondents' attitudes range from 9 to 45, with a cutoff of greater than or equal to 70% (31.5–45) being considered favorable attitudes. In comparison, less than 70% (31.5) will be unfavorable attitude toward the COVID-19 vaccine [34].

The behavioral components (perceived susceptibility to COVID-19, COVID-19 vaccination benefit, and potential vaccine harms) were assessed using three-item questionnaires [35]. These behavioral components were assessed on a 5-point Likert scale from 'Strongly agree' (score 5) to 'Strongly disagree' (score 1) (Strongly Agree, Agree, Neutral, Disagree, Strongly disagree). The sum of each respondent's behavioral characteristics ranges from 3 to 15. The cutoff point was calculated using the demarcation threshold formula: [(highest score-lowest score)/2] + lowest score = [(15–3)/2] +3 = 9 [35–37]. Participants who scored 9 or higher were regarded to have a high perceived susceptibility to COVID-19, perceived COVID-19 vaccination benefit, and perceived potential vaccine harms. In contrast, those who scored less than 9 were considered to have a low perceived susceptibility to COVID-19, perceived COVID-19 vaccination benefit, and perceived potential vaccine harms [35].

## Data collection tool and quality control

The data was collected from eligible study participants using a face-to-face interview technique in the local language (Amharic). The data collectors were eight professional nurses and two general practitioners who oversaw the data gathering under the investigator's direct supervision. To avoid misinterpretation, data collectors were given one-day training. A pre-test involving 5% of the study population was conducted outside the study area, at the Gondar town health center (Poli health facility). Patients with chronic diseases seen in the outpatient department took part in the pre-test. Based on the pre-test results, we made as few adjustments as possible to data collection instruments before the actual data collection. Furthermore, the pre-test results were always utilized to assess the validity and reliability of the data collection instrument and define appropriate data collection techniques [38]. Cronbach's alpha results from the pre-test were used to determine the internal consistency of each dimension of the data collection instrument. Scores on COVID-19 vaccine knowledge (Cronbach alpha = 0.78), COVID-19 vaccine attitudes (Cronbach alpha = 0.92), perceived susceptibility to COVID-19 (Cronbach alpha = 0.77), perceived COVID-19 vaccination benefit (Cronbach alpha = 0.75) and perceived potential vaccine harms (Cronbach alpha = 0.76) were on the acceptable range.

## Data processing and analysis

Once all necessary data are obtained, data will be checked for completeness, edited, cleaned, coded, and entered into Epi-data version 4.6 and analyzed by SPSS version 25. Descriptive statistics were computed to describe the socio-demographic variables and chronic patients' willingness to take COVID-19 Vaccine. Bivariable and multivariable binary logistic regression models were used to determine the relationship between the dependent and independent variables. Variables with a p-value of less than 0.2 in the bivariable analysis were considered candidate variables for the multivariable logistic regression analysis. Then, variables that show statistically significant association with a p-value less than 0.05 in the multivariable logistic regression analysis will be considered predictors for willingness to take COVID-19 vaccines. Multi-collinearity assumptions were validated before running the logistic regression model. All of the variance inflation factor (VIF) values were less than three, indicating that multi-collinearity was not present.

## Ethical approval and consent to participate

We confirm that all methods were carried out in accordance with the principles of the Helsinki Declaration. The institutional review committee (IRC) of the school of medicine, University of Gondar provided ethical clearance and permission letters. After being informed about the study's purpose, each patient signed a consent form. The University of Gondar's specialized

hospital provided a letter of support. By keeping participants anonymous, privacy and confidentiality were ensured during data collection. Chronic patients' participation was entirely voluntary, and they were free to leave the study at any time if they were dissatisfied with the survey.

## Results

### Socio-demographic characteristics and experience of vaccine

A total of 401 chronic patients participated in this study, with a response rate of 94.7%. 19 participants were ruled ineligible for the study because they met the exclusion criteria, and 3 people were refused participation. The mean age of respondents was 45.8 years (SD ± 16), with the age range of 18 to 87 years. Of all respondents, 108(26.9%) were between 50 to 60 years of age, and about half, 206(51%) of the participants were males. The majority, 245(61.1%) of respondents, were urban residents, and about 28.4% were above high school educational levels. Only 49(12.2%) were working health-related jobs of the total participants. The most prevalent chronic cases in the study area were found to be hypertension 95(23.7%) and diabetic Mellitus 85(21.2%) (**Table 1**).

Among the 401 adult chronic patients, 209 (52.1%) had previously received a flu vaccination, and 103 (25.7%) had previously declined vaccination. Of all respondents, 74(18.5%) indicated that they had a previous infection with COVID-19, while 125 (31.5%) said a family member or friend had been infected previously. Approximately 72(18%) of the participants reported the death of a family member or friend due to the COVID-19 complication (**Table 1**).

### Knowledge of the COVID-19 vaccine

Of the total respondents, 140 (34.9%) with [95% CI (30.2–39.6%)] of study participants had Good knowledge about the COVID-19 vaccine (**Fig 1**). As presented in **Table 2**, almost 94% of respondents were aware of the availability of the COVID-19 vaccine in Ethiopia. However, only 212(52.9%) of respondents reported that they had information about the effectiveness of the COVID-19 vaccination. Half of the chronic patients in this study believed that vaccines prevent COVID-19 infection. Around 217(54%) of the total respondents were argued to be at a higher risk of COVID-19 infection than the general population. The majority of participants, 290 (72.3%), were aware of the free COVID-19 vaccination in Ethiopia.

### Attitude towards COVID-19 vaccine

Of the total respondents, 162 (40.4%) with [95% CI (35.5–45.2%)] expressed a Good attitude towards the COVID-19 vaccine (**Fig 1**). **Table 3** shows that 104(25.9%) and 95(23.7%) of the respondents strongly agreed that the COVID-19 vaccination is necessary and safe. Approximately one–third of the respondents believed that patients with chronic conditions should be prioritized in the vaccination program. Only 52 (13.0%) of the total respondents were concerned about the COVID-19 vaccination's side effects, and also 38 (9.5%) believed the COVID-19 vaccine could create long-term health problems. Surprisingly, 58(14.5%) of respondents agreed that they would purchase the vaccine if the government did not supply it free of charge.

### Behavioral characteristics towards COVID-19 vaccine

Respondents' behavioral aspects of the COVID-19 vaccine were evaluated based on the health belief model constructs (perceived susceptibility to COVID-19, vaccine benefit, and

**Table 1. Participants socio-demographic characteristics and their vaccine experience.**

| Socio-demographic characteristics | | Frequency | Percentage |
|---|---|---|---|
| Sex | Male | 206 | 51.4 |
| | Female | 195 | 48.6 |
| Age | less than 30 | 85 | 21.2 |
| | 30–39 | 61 | 15.2 |
| | 40–49 | 80 | 20.0 |
| | 50–60 | 108 | 26.9 |
| | above 60 | 67 | 16.7 |
| Marital status | Single | 107 | 26.7 |
| | Married | 179 | 44.6 |
| | Divorced | 63 | 15.7 |
| | Widowed | 52 | 13.0 |
| Religion | Orthodox Christian | 267 | 66.6 |
| | Muslim | 87 | 21.7 |
| | Protestant | 38 | 9.5 |
| | Others* | 9 | 2.2 |
| Residence | Urban | 245 | 61.1 |
| | Rural | 156 | 38.9 |
| Occupation | Government employ | 59 | 14.7 |
| | Private business | 52 | 13.0 |
| | Housewife | 72 | 18.0 |
| | Merchant | 47 | 11.7 |
| | Daily laborer | 28 | 7.0 |
| | Student | 43 | 10.7 |
| | Farmer | 85 | 21.2 |
| | Not-working | 15 | 3.7 |
| Educational level | Can't read and write | 77 | 19.2 |
| | Can read and write | 85 | 21.2 |
| | Elementary school | 76 | 19.0 |
| | High school | 49 | 12.2 |
| | Above high school class | 114 | 28.4 |
| Healthcare-related workers | No | 352 | 87.8 |
| | Yes | 49 | 12.2 |
| Chronic illness | Diabetes mellitus | 85 | 21.2 |
| | Hypertension | 95 | 23.7 |
| | Cardio vascular disease | 61 | 15.2 |
| | Chronic kidney disease | 41 | 10.2 |
| | Asthma | 31 | 7.7 |
| | HIV/AIDS | 38 | 9.5 |
| | Others** | 50 | 12.5 |
| Have you ever received any flu vaccination before | No | 192 | 47.9 |
| | Yes | 209 | 52.1 |
| Have you ever refused vaccination in the past | No | 298 | 74.3 |
| | Yes | 103 | 25.7 |
| Have you ever been infected by COVID-19 | No | 327 | 81.5 |
| | Yes | 74 | 18.5 |
| Friends or family members infected by COVID-19 | No | 276 | 68.8 |
| | Yes | 125 | 31.2 |

(*Continued*)

**Table 1.** (Continued)

| Socio-demographic characteristics | | Frequency | Percentage |
|---|---|---|---|
| Lost family members or friends due to COVID-19 | No | 329 | 82 |
| | Yes | 72 | 18 |

*Catholic and Adventist

**psychiatric, Arthritis, cancer, interstitial lung diseases, chronic obstructive lung disease, neurodegenerative diseases like Alzheimer and Parkinsons diseases, inflammatory bowel diseases

seriousness) (**Fig 2**). The majority (69%) of chronic patients were perceived to be susceptible to COVID-19 infection. Similarly, About 67% of the respondents perceived benefiting from the COVID-19 vaccine. However, 47.9% of respondents perceived the COVID-19 vaccination effects were substantial.

## Willingness to take the COVID-19 vaccine and their sources of vaccine information

Overall, 219 (54.6%) with [95% CI (49.7–59.5%)] of study participants reported that they are willing to be vaccinated against COVID-19. In contrast, 182 (45.4%) were unwilling to receive the COVID-19 vaccine (**Fig 3**). Among the various electronic health information sources, the mass media was the most important 231(57.6%) source of information about the COVID-19 vaccine for adult chronic patients (**Table 4**). Social media networking sites were also the second most popular 97(24.2%) sources of vaccine information.

## Factors associated with chronic patient's willingness to take COVID-19 vaccination

Bivariate and multivariable analyses were used to analyze the factors influencing chronic patients' willingness to accept the COVID-19 vaccination. In a bivariate analysis, sex, residence, educational level, occupation, being a health worker, Previous flu vaccination, previous COVID-19 infection, loss of family members or friends due to COVID-19, knowledge about

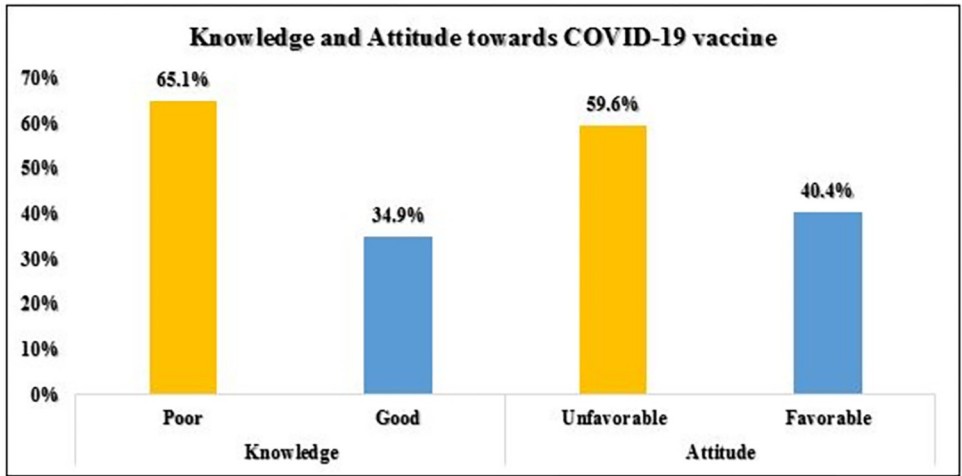

**Fig 1. Respondent's knowledge and attitude towards COVID-19 vaccine.**

**Table 2. Knowledge about the COVID-19 vaccine.**

| Knowledge variables | No N(%) | Yes N(%) |
|---|---|---|
| Do you have information about the presence of the COVID-19 vaccine? | 25(6.2) | 376(93.8) |
| Do you have information about the effectiveness of the COVID-19 vaccine? | 189(47.1) | 212(52.9) |
| Do vaccines prevent COVID-19 infection? | 187(46.6) | 214(53.4) |
| Are you at increased risk of COVID-19 infection than the general population? | 184(45.9) | 217(54.1) |
| Is there a cure for COVID-19 infection? | 199(49.6) | 202(50.4) |
| Is the COVID-19 vaccine provided free of charge in Ethiopia? | 111(27.7) | 290(72.3) |

COVID-19 vaccine, attitude toward COVID-19 vaccine, perceived susceptibility, and perceived benefit were the candidate variables for the multivariable logistic regression analysis.

Based on the results of multivariable logistic regression analysis, Being a healthcare worker (AOR = 2.94, 95% CI: 1.24–6.96), Lost family members or friends due to COVID-19 (AOR = 2.47, 95% CI: 1.21–5.00), good knowledge about COVID-19 vaccine (AOR = 2.44, 95% CI: 1.37–4.33), favorable attitude towards COVID-19 vaccine (AOR = .8.56 95% CI: 4.76–15.38), perceived susceptibility of the COVID-19 infection (AOR = 2.94, 95% CI: 1.62–5.33) and perceived benefit of the COVID-19 vaccine (AOR = 1.89, 95% CI: 1.08–3.31), were found to be significantly associated with willingness to receive the COVID-19 vaccine among chronic patients, as shown in (**Table 5**).

## Discussion

This study investigates the willingness to accept the COVID-19 vaccine and the associated factors among people with chronic illnesses in Northwest Ethiopia. According to the findings, more than half of those with chronic diseases were willing to receive a COVID-19 vaccine. The results also revealed that being a health care professional, lost family or friends due to COVID-19, having good knowledge about the COVID-19 vaccine, a favorable attitude towards COVID-19 vaccine, perceived susceptibility of the COVID-19 infection, and perceived benefit of the COVID-19 vaccination were all associated with higher vaccine willingness.

In this study, 54.6% of adult chronic patients were willing to receive the COVID-19 vaccination. The findings of this study are consistent with prior investigations involving similar study participants, with 52% willing to accept the COVID-19 vaccine in Saudi Arabia [39]. The results of this study are higher than those of a population-based study conducted in Ethiopia, where only 31.4% were willing to get the COVID-19 vaccination [15]. The possible

**Table 3. Attitude towards COVID-19 vaccine.**

| Attitude variables | SD (%) | DA (%) | N (%) | A (%) | SA (%) |
|---|---|---|---|---|---|
| The COVID-19 vaccines are essential for us. | 34(8.5) | 43(10.7) | 62(15.5) | 158(39.4) | 104(25.9) |
| The newly discovered Covid-19 vaccines are safe. | 39(9.7) | 73(18.2) | 97(24.2) | 97(24.2) | 95(23.7) |
| I encourage my family/friends/relatives to receive the vaccine. | 33(8.2) | 46(11.5) | 79(19.7) | 149(37.2) | 94(23.4) |
| People with chronic diseases should be given priority during vaccination. | 22(5.5) | 40(10.0) | 60(15.0) | 163(40.6) | 116(28.9) |
| I am afraid of the side effects of the COVID-19 vaccine. | 57(14.2) | 84(20.9) | 86(21.4) | 122(30.4) | 52(13.0) |
| COVID-19 vaccine may cause long-term health problems for me. | 72(18.0) | 107(26.7) | 98(24.4) | 86(21.4) | 38(9.5) |
| We cannot decrease the frequency of COVID-19 without vaccination. | 43(10.7) | 85(21.2) | 93(23.2) | 131(32.7) | 49(12.2) |
| I will buy the vaccine if the government does not provide it freely? | 49(12.2) | 103(25.7) | 77(19.2) | 114(28.4) | 58(14.5) |
| A COVID-19 vaccine should be compulsory for all patients with chronic diseases? | 30(7.5) | 89(22.2) | 104(25.9) | 116(28.9) | 62(15.5) |

**SD** strongly disagree, **DA** disagree, **N** neutral, **A** agree, **SA** strongly agree

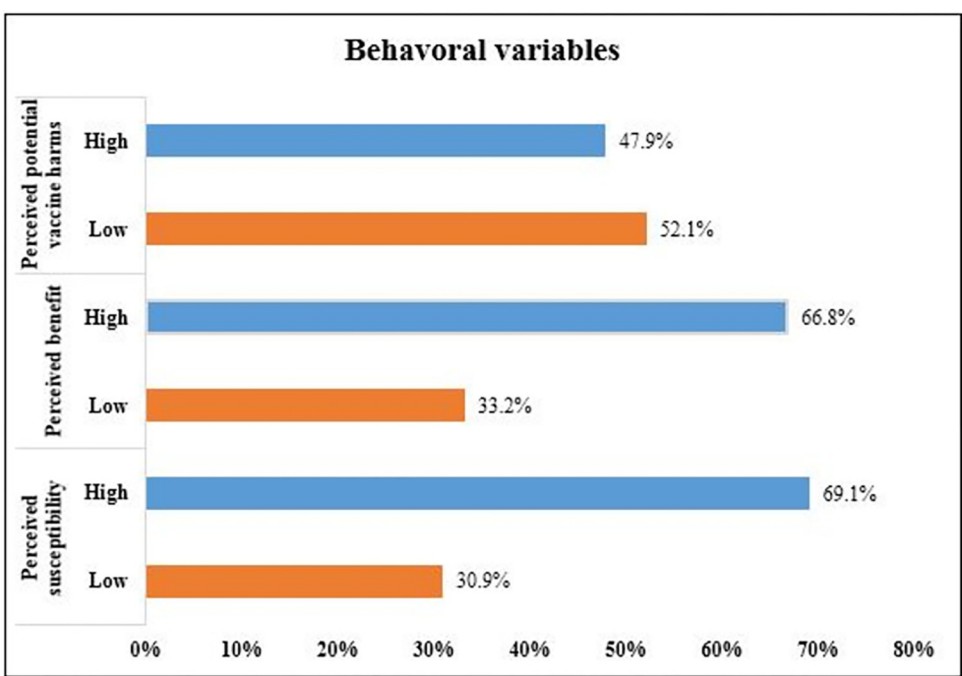

**Fig 2. Respondent's behavioral characteristics toward COVID-19 vaccine.**

explanation for this discrepancy might be that chronic patients have a higher perceived vulner-ability for the COVID-19 infection than the general population [40]. In contrast, the findings of this study are lower than a study conducted in Ethiopia (61%) [41] and (72.2%) [42], India (89.4%) [43], Bangladesh (74.5%) [44]. The disparity could be explained by differences in access to health care services, understanding of the severity of COVID-19, and the difference in the study population. Another possible explanation for the observed difference is the

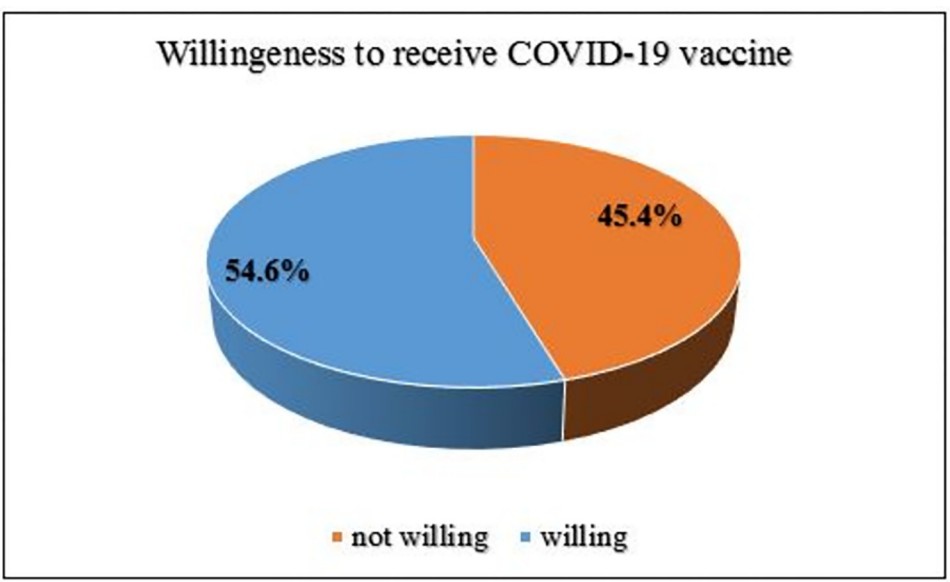

**Fig 3. Chronic patient's willingness to receive COVID-19 vaccine.**

**Table 4. The primary source of information about COVID-19 vaccines.**

| Source of information | Frequency | Percent |
|---|---|---|
| Mass media | 231 | 57.6 |
| Social media networking sites | 97 | 24.2 |
| Religious Personnel | 27 | 6.7 |
| Google | 19 | 4.7 |
| Others* | 27 | 6.7 |

*Friends or family members, Newspaper, Brothers

**Table 5. Factors associated with chronic patients' willingness to take COVID-19 vaccination (N = 401).**

| Characteristics | | Willingness | | COR (CI 95%) | AOR (CI 95%) | P-value |
|---|---|---|---|---|---|---|
| | | Willing (%) | Not willing (%) | | | |
| Sex | Male | 119(29.7) | 87(21.7) | 1.29(0.87–1.92) | 1.25(0.70–2.24) | 0.437 |
| | Female | 100(24.9) | 95(23.7) | 1 | 1 | |
| Residence | Urban | 141(35.2) | 104(25.9) | 0.73(0.49–1.10) | 0.86(0.45–1.63) | 0.656 |
| | Rural | 78(19.5) | 78(19.5) | 1 | 1 | |
| Educational level | Above high school class | 72(18.0) | 42(10.5) | 0.60(0.30–1.19) | 0.76(0.30–1.89) | 0.560 |
| | high school | 25(6.2) | 24(6.0) | 0.55(0.30–0.99) | 0.62(0.24–1.63) | 0.341 |
| | Elementary class | 37(9.2) | 39(9.7) | 0.75(0.42–1.34) | 0.85(0.37–1.95) | 0.701 |
| | Can read and write | 48(12.0) | 37(9.2) | 0.54(0.30–0.97) | 1.17(0.51–2.65) | 0.705 |
| | Can't read and write | 37(9.2) | 40(10.0) | 1 | 1 | |
| Occupation | Government employ | 40(10.0) | 19(4.7) | 1.84(0.58–5.83) | 0.97(0.23–4.05) | 0.977 |
| | Private business | 31(7.7) | 21(5.2) | 1.29(0.40–4.10) | 0.73(0.17–3.03) | 0.673 |
| | Housewife | 36(9.0) | 36(9.0) | 0.87(0.28–2.66) | 0.61(0.14–2.66) | 0.519 |
| | Merchant | 26(6.5) | 21(5.2) | 1.08(0.33–3.47) | 0.68(0.16–2.93) | 0.613 |
| | Daily laborer | 12(3.0) | 16(4.0) | 0.65(0.18–2.31) | 0.33(0.06–1.61) | 0.171 |
| | Student | 23(5.7) | 20(5.0) | 1.06(0.31–3.26) | 0.46(0.10–1.98) | 0.300 |
| | Farmer | 43(10.7) | 42(10.5) | 0.89(0.29–2.69) | 0.49(0.11–2.07) | 0.336 |
| | Not-working | 8(2.0) | 7(1.7) | 1 | 1 | |
| Healthcare worker | Yes | 37(9.2) | 12(3.0) | 2.88(1.45–5.70) | 2.94(1.24–6.96) | 0.014* |
| | No | 182(45.4) | 170(42.4) | 1 | 1 | |
| Received flu vaccination in the past | Yes | 125(31.2) | 84(20.9) | 1.55(1.04–2.30) | 0.95(0.55–1.64) | 0.878 |
| | No | 94(23.4) | 98(24.4) | 1 | 1 | |
| Infected by COVID-19 before | Yes | 48(12.0) | 26(6.5) | 1.68(0.99–2.84) | 0.92(0.46–1.84) | 0.832 |
| | No | 171(42.6) | 156(38.9) | 1 | 1 | |
| Lost family members or friends due to COVID-19 | Yes | 47(11.7) | 25(6.2) | 1.71(1.01–2.91) | 2.47(1.21–5.00) | 0.012* |
| | No | 172(42.9) | 157(39.2) | 1 | 1 | |
| Knowledge | Good | 99(24.7) | 41(10.2) | 2.83(1.83–4.39) | 2.44(1.37–4.33) | 0.001* |
| | Poor | 120(29.9) | 141(35.2) | 1 | 1 | |
| Attitude | Favorable | 138(34.4) | 24(6.0) | 11.21(6.73–18.66) | 8.56(4.76–15.38) | 0.000* |
| | Unfavorable | 81(20.2) | 158(39.4) | 1 | 1 | |
| Perceived susceptibility | High | 188(46.9) | 89(22.2) | 6.33(3.92–10.22) | 2.94(1.62–5.33) | 0.000* |
| | Low | 31(7.7) | 93(23.2) | 1 | 1 | |
| Perceived benefit | High | 176(43.9) | 92(22.9) | 4.00(2.57–6.23) | 1.89(1.08–3.31) | 0.025* |
| | Low | 43(10.7) | 90(22.4) | 1 | 1 | |

widespread dissemination of anti-vaccination misinformation on various social media sites [39, 45].

Health care professionals with chronic conditions were 2.94 times more willing than other adult chronic patients to obtain the COVID-19 vaccination. The possible explanation is that health professionals with chronic diseases have a greater understanding of the risk of COVID-19 infections than other population groups [46]. This conclusion, however, contradicts the earlier research [15]. The disparity may be related to the fact that prior research was conducted among the general population. Therefore, there may be differences in willingness between health professionals with chronic diseases and those who do not have chronic illnesses.

The current study revealed that among people with chronic diseases, those who lost family members or friends due to COVID-19 were 2.47 times more willing to be vaccinated for COVID-19 compared with those who did not lose any friends or family members. This could be related to a high-risk perception following losing family members or friends due to COVID-19 infections [47]. These findings are consistent with other research conducted in Asia, Africa, and Europe [39, 47–49].

Chronic patients who had good knowledge about the COVID-19 vaccine were 2.44 times more willing to receive the COVID-19 vaccination than those who had a poor understanding of the vaccine. The findings were consistent with research conducted in southern Ethiopia [41], Addis Ababa [50], and south-Africa [51]. The possible explanation might be those chronic patients who had good knowledge about the COVID-19 vaccine have better information on the advantages of vaccination. Besides this, this finding is supported by the "Reason Action Theory" which states that an individual's intention to acquire a particular behavior results from their current knowledge regarding that behavior [52].

The attitude variable is significantly associated with the chronic patients' willingness to accept the COVID-19 vaccination. Patients who had a favorable attitude towards the COVID-19 vaccine were 8.56 times more willing to take the COVID-19 vaccination than those who had an unfavorable attitude. This could be due to chronic patients who had a favorable attitude toward the COVID-19 vaccine were being aware of the influence of COVID-19 viruses on community health [42]. As a result, to reduce the risk of COVID-19 complications, they may accept the COVID-19 vaccine. This study's findings contradict prior research conducted in Ethiopia in which attitude is not a significant variable for the COVID-19 vaccination willingness [41]. The disparity could be explained by the fact that COVID-19 complications worsen from time to time, and chronic patients might have a more favorable attitude than in previous studies [53]. The other reason for the difference might be the difference in the study population, in which chronic patients have a greater risk of COVID-19 infection than the other population groups [54].

According to the findings of this study, people with a high perceived susceptibility to COVID-19 infection were 2.5 times more willing to receive the COVID-19 vaccine against the pandemic. This finding was consistent with reports of the previous studies [35, 55, 56]. This could be because if individuals believe they are vulnerable, they will reduce their risk. Furthermore, another factor influencing participants' willingness to accept the COVID-19 vaccine was their perception of its benefits. And the finding is supported by a study conducted in Ethiopia [35] and Japan [57]. This is because individuals' perception of the potential benefit of the vaccination will increase their willingness toward the COVID-19 vaccination [58].

## Conclusion

This study confirms that 54.6% of the respondents were willing to receive the COVID-19 vaccine. Being a health professional, having lost family members or friends due to COVID-19,

having good knowledge about the COVID-19 vaccine, having a favorable attitude toward the COVID-19 vaccine, having a high perceived susceptibility to COVID-19 infection, and having a high perceived benefit of the COVID-19 vaccination were all significant variables in adult chronic patients' willingness to receive COVID-19 vaccination. As a result, providing health education for chronic patients to emphasize the knowledge and attitude of the COVID-19 vaccine and raise patients' perceived risk of COVID-19 and the benefit of the COVID-19 vaccine could be recommended to improve their acceptance of COVID-19 vaccination.

## Strength and limitations

The findings from this study provide valuable information on the willingness to receive the COVID-19 vaccination on adult chronic patients in resource-limited settings. The limitation of this study is that because it was an institution-based cross-sectional survey, only health professionals who arrived during the data collection period were interviewed.

## Supporting information

**S1 Data.**
(SAV)

## Acknowledgments

We are incredibly grateful to the University of Gondar School of Medicine for allowing us to conduct this research. Our gratitude also goes to the managing board of the University of Gondar specialized hospital for providing all necessary information and assistance. We would also like to express our appreciation for the study participants, data collectors, and supervisors.

## Author Contributions

**Conceptualization:** Masresha Derese Tegegne, Mehretie Kokeb.

**Data curation:** Masresha Derese Tegegne, Surafel Girma, Surafel Mengistu, Tadele Mesfin, Tenanew Adugna.

**Formal analysis:** Masresha Derese Tegegne, Mehretie Kokeb.

**Investigation:** Masresha Derese Tegegne.

**Methodology:** Masresha Derese Tegegne.

**Software:** Masresha Derese Tegegne.

**Supervision:** Masresha Derese Tegegne, Mehretie Kokeb.

**Validation:** Masresha Derese Tegegne.

**Visualization:** Masresha Derese Tegegne, Mehretie Kokeb.

**Writing – original draft:** Masresha Derese Tegegne, Surafel Girma, Surafel Mengistu, Tadele Mesfin, Tenanew Adugna.

**Writing – review & editing:** Mehretie Kokeb, Endalkachew Belayneh Melese, Yilkal Belete Worku, Sisay Maru Wubante.

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
