## [Decision Letter · Decision Letter 0]

4 May 2022

PONE-D-22-03149Willingness to receive COVID-19 vaccine and associated factors among adult chronic patients. A cross-sectional study in Northwest EthiopiaPLOS ONE

Dear Dr. Tegegne,

Thank you for submitting your manuscript to PLOS ONE. After careful consideration, we feel that it has merit but does not fully meet PLOS ONE’s publication criteria as it currently stands. Therefore, we invite you to submit a revised version of the manuscript that addresses the points raised during the review process.

We look forward to receiving your revised manuscript.

Kind regards,

Khin Thet Wai, MBBS, MPH, MA

Academic Editor

PLOS ONE

Journal Requirements:

Additional Editor Comments (if provided):

English language correction by the professional scientific language editing service is essential.

Reviewers' comments:

Reviewer's Responses to Questions

**Comments to the Author**

1. Is the manuscript technically sound, and do the data support the conclusions?

Reviewer #1: Partly

Reviewer #2: Yes

2. Has the statistical analysis been performed appropriately and rigorously? 

Reviewer #1: Yes

Reviewer #2: Yes

3. Have the authors made all data underlying the findings in their manuscript fully available?

Reviewer #1: No

Reviewer #2: Yes

4. Is the manuscript presented in an intelligible fashion and written in standard English?

Reviewer #1: No

Reviewer #2: Yes

5. Review Comments to the Author

Reviewer #1: I was invited to review the paper entitled:” Willingness to receive COVID-19 vaccine and associated factors among adult chronic patients. A cross-sectional study in Northwest Ethiopia”. The study aimed to investigate willingness to receive COVID-19 vaccination and factors affected it among adult with chronic diseases. There has been a great deal of research in this area internationally, and I did not notice any novelty in the results of this study from an international viewpoint. There are also several concerns with this study.

1. Throughout the Background section, there are many inappropriate citations. For example, the authors cite reference number [9] for the history of vaccine development. However, this is a cross-sectional study to clarify vaccine hesitancy in Spain and is not an appropriate citation. The same thing happened with reference numbers [10], [11]. Reference number [16] is also inappropriate for citation because it is about vaccine hesitancy before COVID-19. The author should research previously research, and provide appropriate citations.

2. The sampling method of the target participants is very unclear.

For example

What exactly is a chronic disease?

Who diagnosed the chronic diseases and how?

Who made the decision to exclude the patients?

What was the total number of patients who met the inclusion and exclusion criteria?

How were the participants randomly selected?

These points are not described in the current manuscript, and reproducibility is poor. At least in the current description, it cannot be called “random sampling technique”. Moreover, I am concerned about the possibility of a large selection bias in this paper.

3. The validity and reliability of the questionnaire (or survey item) is unknown.

The authors described the translation procedure of the questionnaire, but there is no description of the validity and reliability of the original questionnaire.

Also, the translation procedure is not sufficiently described. (Reference: https://pubmed.ncbi.nlm.nih.gov/15804318/)

4. Looking at Table 1, chronic diseases do not include important chronic diseases, such as cancer and COPD. (Reference: https://www.cdc.gov/coronavirus/2019-ncov/science/science-briefs/underlying-evidence-table.html). I believe the point of this study is that it targeted patients with chronic diseases. Therefore, what is defined as chronic disease and how it is assessed are important points of this study. These should be clearly stated in the Methods section and the Results section.

5. The author described that a self-administered questionnaire was used in abstract and methods section. However, the authors state in the Limitation section that they conducted an Interview, which is inconsistent.

Improvement of these above points is essential to consider the quality and importance of this paper.

The reviewer reserves other comments at this moment.

Reviewer #2: Authors can consider the following suggestions:1.Should update information. For instance, the total number of infected patients and death tolls caused by the pandemic. The authors have used old data in the introduction.

2.Rationale for conducting this study on ‘patients suffering chronic diseases’ should be justified with the help of statistics and existing literature.

3.It seems, the study was conducted before the planned vaccination campaign (end of 2021). That said, readers do not know what happened when the vaccination campaign actually started! Thus, the authors should be informed about the updated situation with reference to other parts of the world.

4.It is good that the authors have explained the inclusion criteria, but it is not clear if they included any minor participants (aged less than 18 years). Also, the exclusion criteria require justification – why those groups of patients were excluded.

5. The manuscript should be thoroughly checked for grammatical accuracy.

6.The study shows that over 54% of participants were willing to receive the vaccine. Well, did the participants mention and/or do the authors have any information on why they were not willing to receive the vaccine? It would be good to know why they were not willing to receive the vaccine.

7.The authors mentioned the proportion of the respondents willing to receive the vaccine. But were they informed about their comorbidities and possible side effects of receiving the vaccine?

6. PLOS authors have the option to publish the peer review history of their article (what does this mean?). If published, this will include your full peer review and any attached files.

Reviewer #1: No

Reviewer #2: **Yes: **Abu-Hena Mostofa Kamal

---

## [Author Response · Author response to Decision Letter 0]

19 May 2022

Point-by-point response letter

Title: Willingness to receive COVID-19 vaccine and associated factors among adult chronic patients. A cross-sectional study in Northwest Ethiopia

Dear, Editors of PLOS ONE

I want to convey our gratitude for reviewing our research on behalf of all authors.We appreciate your time and effort and that of the associate editors and reviewers in offering feedback on our paper. We value your and the reviewers' thoughtful comments and significant improvements to our manuscript. We tried to address the editor's and reviewers' issues and suggestions. The following is a detailed response to the reviewer's suggestions and comments.

Finally, we certify that this paper has not been previously published and is not currently being considered by another publication. All authors have approved the manuscript and agreed to resubmit it to PLOS ONE.

With kind regards!

On behalf of the co-authors

Masresha Derese Tegegne, [MPH in Health informatics] 

Response to editor’s 

We have edited and amended the manuscript based on your suggestions and comments. We appreciate your constructive criticism and suggestions for enhancing this work. Please find a detailed response to the editor's analysis and recommendations below.

Editor comments 

https://journals.plos.org/plosone/s/file?id=wjVg/PLOSOne_formatting_sample_main_body.pdf andhttps://journals.plos.org/plosone/s/file?id=ba62/PLOSOne_formatting_sample_title_authors_affiliations.pdf

Replay#1: Thanks for the notifications; we confirm that our manuscript meets the PLoS one style requirements. 

Replay#2: Thank you for your constructive suggestions. As we explained in the Ethics approval and consent to participate section, each patient signed a consent form after being informed about the study's purpose. This is stated explicitly in the manuscript and submission form.

Replay#3: Thank you for informing us. All of the data created or analyzed during this investigation is contained in this version of the paper and the supplementary file.

Response to reviewer(s') comments 

On behalf of all authors, I'd like to thank the reviewers for their constructive suggestions, which helped to improve the paper's scientific foundation. The criticism was well-received, and most of the reviewers' suggestions were enhanced. Throughout the document in the tracked change file, certain modifications are highlighted. Please see the blue text below for a point-by-point response to the reviewers' complaints and proposals.

Reviewer #1 Evaluation

General Comments 

“I was invited to review the paper entitled:” Willingness to receive COVID-19 vaccine and associated factors among adult chronic patients. A cross-sectional study in Northwest Ethiopia”. The study aimed to investigate willingness to receive COVID-19 vaccination and factors affected it among adult with chronic diseases. There has been a great deal of research in this area internationally, and I did not notice any novelty in the results of this study from an international viewpoint”.

Replay: Dear reviewer, thank you so much for taking the time to review our work! 

Even though there are other international studies on the topic, this one is unique in that it concentrates on chronic patients. Chronic patients were more likely than other population groups to get COVID-19 infection. As a result, knowing their willingness will provide important data for vaccine camping in the research area. Aside from that, there is little evidence conducted on chronic patients in the study area. In general, knowing chronic patients' desire to receive the COVID-19 vaccine and associated factors is the best evidence at this moment.

“There are also several concerns with this study”.

1. Throughout the Background section, there are many inappropriate citations. For example, the authors cite reference number [9] for the history of vaccine development. However, this is a cross-sectional study to clarify vaccine hesitancy in Spain and is not an appropriate citation. The same thing happened with reference numbers [10], [11]. Reference number [16] is also inappropriate for citation because it is about vaccine hesitancy before COVID-19. The author should research previously research, and provide appropriate citations.

Reply#1. Thank you for the valid comments. This will significantly improve the quality of our research. We made significant changes to all of the referenced sources, and we certify that we cited the original reference that reported the notion in this version of the manuscript. Please see the first page of the introduction, lines number, 11,14,17,30 and the second page of the introduction, lines number 2 for your recommendations.

1. The sampling method of the target participants is very unclear.

For example

1.1. What exactly is a chronic disease?

Reply#1.1: Thank you for your feedback; in this revised manuscript, we have explained that “Patients with chronic diseases such as chronic liver disease, chronic kidney disease, hypertension, diabetes mellitus, and heart failure were included in this study”. Please have a look at the Study populations and eligibility criteria section on lines number 7, and 8 for further information.

1.2. Who diagnosed the chronic diseases and how?

Reply#1.2: Thank you for your informative feedback: Chronic disease was diagnosed using a set of clinical features, laboratory findings, and imaging results. Furthermore, when serum IgG levels are relatively high, it is referred to as chronic illness. The treating physicians determined the diagnosis based on the criteria given above. This study comprised patients who had their follow-up at the University of Gondar specialized hospital outpatient department. Please see the details in the study populations and eligibility criteria section on lines number 8,9 and 10.

1.3. Who made the decision to exclude the patients?

Reply#1.3: Thank you for your valuable input; it significantly improves the quality of our paper. Before data collection, one day of training was given to the two supervisors and eight data collectors who were involved in the data collection procedure. As a result, supervisors and data collectors under the strict supervision of the investigator made the decision to omit the participants based on the information supplied in the inclusion and exclusion criteria. Please see line number 9-11 for the data collection tool and quality control section.

1.4. What was the total number of patients who met the inclusion and exclusion criteria?

Reply#1.4: Thanks for your comments: As highlighted in study design and setting section on line number 30 the university of Gondar’s specialized hospital outpatient department serves approximately 6000 chronically ill patients who are followed up on a regular basis based on their unique diseases and severity of illness.

1.5. How were the participants randomly selected?

These points are not described in the current manuscript, and reproducibility is poor. At least in the current description, it cannot be called “random sampling technique”. Moreover, I am concerned about the possibility of a large selection bias in this paper.

Reply#1.5 Thank you for your feedback: Your suggestion is helpful: 

To select study participants, a systematic random sampling process was applied. Every day, an average of 150 chronic patients visit the outpatient departments of the University of Gondar specialized Hospital. A total of 2,250 chronic patients were predicted during the data collection period (15 working days). By dividing the total population by sample size (423), the sample interval was 5. In the first interval, the lottery method was used, and in the subsequent intervals, every fifth chronic patient was chosen until we had a total of 423 samples. Please have a look at the sampling procedure section.

2. The validity and reliability of the questionnaire (or survey item) is unknown.

The authors described the translation procedure of the questionnaire, but there is no description of the validity and reliability of the original questionnaire.

Also, the translation procedure is not sufficiently described. (Reference: https://pubmed.ncbi.nlm.nih.gov/15804318/)

 Reply#2: Thank you for the valid comments, we would like to thank the reviewers for providing the right evidence on the translational procedure. As highlighted in the data collection and quality control section on lines number 13-20, the reliability and validity of the all-composite variables used in the study were determined by using the Cronbach alpha results from the pre-test. The Cronbach alpha scores on COVID-19 vaccine knowledge (Cronbach alpha=0.78), COVID-19 vaccine attitudes (Cronbach alpha=0.92), perceived susceptibility to COVID-19 (Cronbach alpha=0.77), perceived COVID-19 vaccination benefit (Cronbach alpha=0.75) and perceived potential vaccine harms (Cronbach alpha=0.76) were on the acceptable range. Besides this, as indicated in the data collection tool and quality control section on lines number 12,13, best practice was considered by evaluating the pre-test results and by describing the translation procedure of the questionnaire.

3. Looking at Table 1, chronic diseases do not include important chronic diseases, such as cancer and COPD. (Reference: https://www.cdc.gov/coronavirus/2019-ncov/science/science-briefs/underlying-evidence-table.html). I believe the point of this study is that it targeted patients with chronic diseases. Therefore, what is defined as chronic disease and how it is assessed are important points of this study. These should be clearly stated in the Methods section and the Results section.

Reply#3: We found your comments extremely helpful, and thanks for your comments. We didn't forget about the important chronic patients you mentioned; this is a writing issue that we addressed in Table 1 under Chronic illness [other categories]. Patients with psychiatric, arthritis, cancer, interstitial lung disease, chronic obstructive pulmonary disease, neurodegenerative disorders like Alzheimer's and Parkinson's, and inflammatory bowel diseases were studied. For more information, please see the footnote in Table 1.

4. The author described that a self-administered questionnaire was used in abstract and methods section. However, the authors state in the Limitation section that they conducted an Interview, which is inconsistent. Improvement of these above points is essential to consider the quality and importance of this paper. The reviewer reserves other comments at this moment.

Reply#4: Thanks for the valid insights: We confirmed that this was a typing mistake, and the authors performed face-to-face interviews with study participants who were eligible using a validated data collection tool. Please see line number 8 in the data collection tool and quality control section, as well as the abstract section.

Reviewer #2 Evaluation

Authors can consider the following suggestions:

1. Should update information. For instance, the total number of infected patients and death tolls caused by the pandemic. The authors have used old data in the introduction.

Reply#1. Thank you for your feedback. This is crucial to the study's credibility. We used the current prevalence of COVID-19 cases and deaths in accordance with your ideas. Please review the first parageraph on introductions.

2. Rationale for conducting this study on ‘patients suffering chronic diseases’ should be justified with the help of statistics and existing literature.

Reply#2. I appreciate your suggestion. This will be extremely beneficial to the quality of our research. We attempted to justify the significance of the study on chronic patients, as mentioned starting from line number 27, on the first page of the introductory section by using existing evidence. “Even while everyone benefits from the vaccination, not everyone benefits equally. Those at a higher risk of contracting COVID-19 infection, more likely to get very unwell once infected, or at a higher risk of developing complications and dying from the disease will benefit the most from vaccination [18, 19]. Patients with chronic conditions such as diabetes, hypertension, cardiovascular, renal, and cerebrovascular diseases were given priority for vaccination since they were more likely to be affected by this virus and have a significant clinical burden from COVID-19 [18, 20]. ”. Aside from that, the importance of this study was stated on the second page of the introductory section, specifically lines 12-19.

3. It seems, the study was conducted before the planned vaccination campaign (end of 2021). That said, readers do not know what happened when the vaccination campaign actually started! Thus, the authors should be informed about the updated situation with reference to other parts of the world.

Reply#3. Thank you for your remarks, which we found incredibly helpful. We attempted to articulate the vaccination campaign in many countries throughout the world, as highlighted on the second page of the introduction section on lines 3-11.

4. It is good that the authors have explained the inclusion criteria, but it is not clear if they included any minor participants (aged less than 18 years). Also, the exclusion criteria require justification – why those groups of patients were excluded.

Reply#4. We found your comments extremely helpful, and thanks for your comments. In the study population and eligibility criteria section, we explained our inclusion criteria in detail on lines 7-12. Aside from this, as the title suggests, the research focuses on adult chronic patients. As a result, in the Ethiopian context, adult refers to a group of persons over the age of 18, hence the study automatically excludes those under the age of 18. Furthermore, we excluded patients who were mentally ill, unable to talk, or who had a significant hearing loss since they were unable to respond to questions about their willingness and permission. Please review the study population and eligibility criteria section, lines 12-15.

5. The manuscript should be thoroughly checked for grammatical accuracy.

Reply#5. Thank you very much for forwarding your comments. We certified that we attempted to improve the writing in this version of the manuscript. Please check over the entire document in the tracked change file; the work was reviewed for improved presentation by the writers and experienced English language specialists.

6. The study shows that over 54% of participants were willing to receive the vaccine. Well, did the participants mention and/or do the authors have any information on why they were not willing to receive the vaccine? It would be good to know why they were not willing to receive the vaccine.

Reply#6. Thank you for the comment! In addition to giving descriptive findings, this study focuses on identifying possible factors and their strength of association with the willingness. The descriptive findings and their regression associations of identified factors in prior investigations were taken into account. As a result, we examined a variety of factors, including perceived possible vaccine harm, lack of knowledge, and a negative attitude toward the COVID-19 vaccine. Please have a look at the result section.

7. The authors mentioned the proportion of the respondents willing to receive the vaccine. But were they informed about their comorbidities and possible side effects of receiving the vaccine?

Reply#7. Thank you for taking the time to share your thoughts. During the research period, the Ethiopian public health institute and the ministry of health enhanced their awareness. The Ethiopian Public Health Institute sends out two short messages about COVID-19, comorbidities, and possible adverse effects, as well as a helpline from the Ministry of Health.

---

## [Decision Letter · Decision Letter 1]

24 May 2022

PONE-D-22-03149R1Willingness to receive COVID-19 vaccine and associated factors among adult chronic patients. A cross-sectional study in Northwest EthiopiaPLOS ONE

Dear Dr. Tegegne,

Thank you for submitting your manuscript to PLOS ONE. After careful consideration, we feel that it has merit but does not fully meet PLOS ONE’s publication criteria as it currently stands. Therefore, we invite you to submit a revised version of the manuscript that addresses the points raised during the review process.

We look forward to receiving your revised manuscript.

Kind regards,

Khin Thet Wai, MBBS, MPH, MA

Academic Editor

PLOS ONE

Journal Requirements:

Additional Editor Comments:

English language correction is deemed necessary

Reviewers' comments:

Reviewer's Responses to Questions

**Comments to the Author**

1. If the authors have adequately addressed your comments raised in a previous round of review and you feel that this manuscript is now acceptable for publication, you may indicate that here to bypass the “Comments to the Author” section, enter your conflict of interest statement in the “Confidential to Editor” section, and submit your "Accept" recommendation.

Reviewer #1: (No Response)

2. Is the manuscript technically sound, and do the data support the conclusions?

Reviewer #1: Yes

3. Has the statistical analysis been performed appropriately and rigorously? 

Reviewer #1: Yes

4. Have the authors made all data underlying the findings in their manuscript fully available?

Reviewer #1: Yes

5. Is the manuscript presented in an intelligible fashion and written in standard English?

Reviewer #1: Yes

6. Review Comments to the Author

Reviewer #1: I confirm that the revision has greatly improved this manuscript.

However, there are still several points that need to be improved.

Background section

1. The authors cited Wikipedia as the source for the description of COVAX (Reference number: 10), but I think that's not adequate. I believe that the information on Wikipedia is inaccurate because an unspecified number of users can freely change the description. Please consider citing a official website, such as WHO website, or article about the COVAX description.

Methods section

2. Please describe the pre-test in more detail. Who was the target participant, and how many were eligible, etc.

3. The authors cited recommendations by the ISPOR task force in their description about the pre-test (Reference number: 40). But it is not clear what exactly they did. For example, did the authors modify the translated text based on the pre-test results, or did they just do cognitive debriefing and proofreading? Please describe in more detail what the authors did specifically based on the results of the pre-test.

4. I understood that the authors evaluated all indicators using interview method. However, I think the current description in methods and abstract section do not clearly state that they have been evaluated in interviews. In addition, there is still the description " questionnaire" in part of the methods section, which is confusing. Please make these points clearer.

Result section

5. For the 22 excluded persons out of 423, please describe the reasons for their exclusion in the results section. How many refused to cooperate with the study? How many met the exclusion criteria such as hearing impairment?

Reference

6. I think some of the descriptions in the Reference are incorrect. For example, Reference number 8 is probably from a web page, but the URL is not listed. I think the description "Organization WH" is also inappropriate. Please double check all the descriptions in the Reference list.

7. PLOS authors have the option to publish the peer review history of their article (what does this mean?). If published, this will include your full peer review and any attached files.

Reviewer #1: No

---

## [Author Response · Author response to Decision Letter 1]

28 May 2022

Point-by-point response letter

Title: Willingness to receive COVID-19 vaccine and associated factors among adult chronic patients. A cross-sectional study in Northwest Ethiopia

Dear, Editors of PLOS ONE

I want to express our gratitude on behalf of all of the authors for taking the time to read our paper, "Willingness to receive COVID-19 vaccine and associated factors among adult chronic patients. A cross-sectional study in Northwest Ethiopia ".

We appreciate your time and work in providing input on our manuscript for the second time and those of the associate editors and reviewers. We appreciate your careful remarks and significant improvements to our work and those of the reviewers. We attempted to resolve the issues and ideas raised by the editor and reviewers. The following is a comprehensive response to the reviewer's opinions and criticisms.

Finally, we certify that this paper has not been previously published and is not currently being considered by another publication. All authors have approved the manuscript, and they have agreed to resubmit it to PLOS ONE Journal.

With kind regards!

On behalf of the co-authors

Masresha Derese Tegegne, [MPH in Health informatics] 

Response to editor’s 

We appreciate your constructive criticism and suggestions for enhancing this work. We have edited and amended the manuscript based on your suggestions and comments. Please find a detailed response to the editor's analysis and recommendations below. 

Editor comments 

Replay#1: Thank you for the notices; we certify that we critically checked our references and that no references have been retracted. 

2. English language correction is deemed necessary

Replay#2: Thank you very much for forwarding your comments. We certified that we attempted to improve the writing in this version of the manuscript. Please check over the entire document in the tracked change file; the work was reviewed for improved presentation by the writers and experienced English language specialists.

Response to reviewer(s') comments 

Dear Reviewer#1: We appreciate your valuable suggestions for the second time, which are crucial in refining the paper's scientific foundation. Most of your requests were improved as a result of the feedback. In the tracked change file, some changes are highlighted throughout the document. Please see the complete responses below for a thorough response to your complaints and recommendations.

 Reviewer #1 Evaluation

General Comments 

 “I confirm that the revision has greatly improved this manuscript.”

Replay#: Thank you for your kind words; your suggestions were valuable, and we will use them to improve the scientific foundations of our research.

“However, there are still several points that need to be improved”.

Background section

1. The authors cited Wikipedia as the source for the description of COVAX (Reference number: 10), but I think that's not adequate. I believe that the information on Wikipedia is inaccurate because an unspecified number of users can freely change the description. Please consider citing a official website, such as WHO website, or article about the COVAX description.

Replay#1: We appreciate your comments; your concerns have been considered, and we have cited the official websites indicated on the WHO web pages. Please see line 14 in the introduction section for more information.

Methods section

2. Please describe the pre-test in more detail. Who was the target participant, and how many were eligible, etc.

Replay#2: Thanks for your suggestions, as it is highlighted in the data collection tool and quality control section; "A pre-test involving 5% of the study population was conducted outside of the study area, at the Gondar town health center (Poli health center). Patients with chronic diseases in the outpatient department took part in the pre-test". Please see the data collection and quality control section on line number 11-13 for further details.

3. The authors cited recommendations by the ISPOR task force in their description about the pre-test (Reference number: 40). But it is not clear what exactly they did. For example, did the authors modify the translated text based on the pre-test results, or did they just do cognitive debriefing and proofreading? Please describe in more detail what the authors did specifically based on the results of the pre-test.

Replay#3: Thank you for your useful feedback; “Based on the pre-test results, we made as few adjustments as possible to data collection instruments before the actual data collection”. Please see the data collection and quality control section on line number 14.

4. I understood that the authors evaluated all indicators using interview method. However, I think the current description in methods and abstract section do not clearly state that they have been evaluated in interviews. In addition, there is still the description " questionnaire" in part of the methods section, which is confusing. Please make these points clearer.

Replay#4: Thank you for your well-considered remarks. "Data was collected from eligible study participants using a face-to-face interview technique in the local language (Amharic)." Please look at lines 9 and 8, respectively, for the abstract and method section.

Result section

5. For the 22 excluded persons out of 423, please describe the reasons for their exclusion in the results section. How many refused to cooperate with the study? How many met the exclusion criteria such as hearing impairment?

Replay#5: Thank you for your feedback; As highlighted in lines number 15-17 in the result section, 19 people were judged ineligible for the study because they met the exclusion criteria, and 3 people were refused participation.

Reference

6. I think some of the descriptions in the Reference are incorrect. For example, Reference number 8 is probably from a web page, but the URL is not listed. I think the description "Organization WH" is also inappropriate. Please double check all the descriptions in the Reference list.

Replay#5: Thank you for your detailed remarks; please see the reference section for answers to all of your questions. Please see reference number 7 for more information.

---

## [Decision Letter · Decision Letter 2]

1 Jun 2022

Willingness to receive COVID-19 vaccine and associated factors among adult chronic patients. A cross-sectional study in Northwest Ethiopia

PONE-D-22-03149R2

Dear Dr. Tegegne,

We’re pleased to inform you that your manuscript has been judged scientifically suitable for publication and will be formally accepted for publication once it meets all outstanding technical requirements.

Kind regards,

Khin Thet Wai, MBBS, MPH, MA

Academic Editor

PLOS ONE

Additional Editor Comments (optional):

Reviewers' comments:

Reviewer's Responses to Questions

**Comments to the Author**

1. If the authors have adequately addressed your comments raised in a previous round of review and you feel that this manuscript is now acceptable for publication, you may indicate that here to bypass the “Comments to the Author” section, enter your conflict of interest statement in the “Confidential to Editor” section, and submit your "Accept" recommendation.

Reviewer #1: All comments have been addressed

2. Is the manuscript technically sound, and do the data support the conclusions?

Reviewer #1: Yes

3. Has the statistical analysis been performed appropriately and rigorously? 

Reviewer #1: Yes

4. Have the authors made all data underlying the findings in their manuscript fully available?

Reviewer #1: Yes

5. Is the manuscript presented in an intelligible fashion and written in standard English?

Reviewer #1: Yes

6. Review Comments to the Author

Reviewer #1: (No Response)

7. PLOS authors have the option to publish the peer review history of their article (what does this mean?). If published, this will include your full peer review and any attached files.

Reviewer #1: No

---

## [Editor Report · Acceptance letter]

4 Jul 2022

PONE-D-22-03149R2 

Willingness to receive COVID-19 vaccine and associated factors among adult chronic patients. A cross-sectional study in Northwest Ethiopia 

Dear Dr. Tegegne:

I'm pleased to inform you that your manuscript has been deemed suitable for publication in PLOS ONE. Congratulations! Your manuscript is now with our production department. 

Kind regards, 

on behalf of

Dr. Khin Thet Wai 

Academic Editor

PLOS ONE